# Evaluation of the Health Promoting Schools (CEPS) Program in the Balearic Islands, Spain

**DOI:** 10.3390/ijerph191710756

**Published:** 2022-08-29

**Authors:** Maria Ramos, Elena Tejera, Elena Cabeza

**Affiliations:** 1Balearic Islands Public Health Department, 07010 Palma, Spain; 2Balearic Islands Health Research Institute (IdisBa), 07120 Palma, Spain

**Keywords:** evaluation, health promotion, education, equity

## Abstract

Objective: To evaluate the structure, process, and results obtained by the CEPS Program. Methods: We combined quantitative and qualitative methods. We reviewed our databases, the health projects and reports sent by the schools, and the notes taken during the assessment visits to the schools. We included all the schools participating in the CEPS Program between 2014–2015 and 2018–2019 for structure purposes, but only those participating for at least two years for the process and results. We used a descriptive analysis as well as a content analysis. Results: 84 schools participated in the CEPS Program. Attrition (24%) occurred mainly after the first year. Most schools performed at least one situation analysis (88.1%) and had at least one teacher trained (73.8%). One of three obtained a certificate (35.7%) or grant (38.1%). For the process and results, we selected 44 schools. Teachers’ participation stood out (61.4% transformative and 38% representative) over the participation of other agents. The coordination of the health committee with other committees and with other local resources was a crucial element. Of the schools, 40.9% had high quality projects, 50% medium, and 9.1% low. Of the schools, 40.9% showed positive health results. Conclusion: We identified the characteristics of high-quality health-promoting schools.

## 1. Introduction

The Health Promoting Schools Program (in Catalan: Centres Educatius Promotors de la Salut [CEPS]) [1] consists of the integration of health promotion in schools through a health project, with the participation of the entire educational community (teachers, non-teaching staff, students, and families), to improve their health and wellbeing as well as the health and wellbeing of the communities where they live and where the schools are. It is based on addressing at least one of the five thematic areas that are related to health and welfare (healthy food and physical activity, emotional wellbeing, sexual and affective education, prevention of addictions, and security and risk prevention) acting simultaneously in six intervention areas (school policies, physical environment, social environment, individual health skills, school team, and links with community). Therefore, although health is not a curricular subject in Spanish schools, in the CEPS Program, health issues are developed transversally in different subjects and also included in school policies, in physical and social environments, and in work with the school staff and other agents in the community.

The CEPS Program follows the model proposed by the World Health Organization [2], by Schools for Health in Europe (SHE) [3], and the theoretical framework of the Ottawa Charter [4], as the CEPS intervention areas correspond to the Ottawa action lines.

The CEPS Program has been running in the Balearic Islands (Mallorca, Menorca, Ibiza, and Formentera) since the 2014–2015 school year, launched by the Public Health Department (PHD) and the Innovation and Educational Community Department (IECD) of the Balearic Islands. It was born from the shared concern about how schools addressed health issues, that is, with isolated activities, to integrate them through a health project after performing a situation analysis. Every public or subsidized school offering infant, primary, secondary, or adult education (basic or secondary education for people over 18 years old) can voluntarily participate. At the beginning of every school year, the IECD launches a call for the CEPS Program. The schools participating must create a health committee, including representatives of the entire educational community, and elect a health coordinator. They are encouraged to carry out a situation analysis using an online tool. Then, they must write a health project for the next two school years, send an annual report of activities, and evaluate their health project after the two school years. We offer training and assessments to the schools. The schools that remain for at least two school years receive a certificate if they request it but only if they have sent the health project and an annual report of activities. The PHD also offers grants of up to 2000 euros for CEPS activities in an annual call [1].

After five school years, it was considered appropriate to evaluate the program. Therefore, the aim was to evaluate the structure, process, and results obtained by the CEPS Program to identify elements that helped or made difficult its development and make recommendations that could help both the schools and PHD and IECD to improve.

## 2. Materials and Methods

We planned the evaluation of the CEPS Program in 2019. The Balearic Islands had in this year 464 schools: 379 public (35 for infants, 272 for infants and primary education, 61 for secondary, and 11 for adults) and 85 subsidized, offering infant, primary, and secondary education.

We included all the schools in the Balearic Islands participating in the CEPS Program between 2014–2015 and 2018–2019 for structure purposes but only those that completed at least two full school years in the CEPS program for process and result evaluation.

We used a mixed design, combining quantitative and qualitative methods, as experts recommend [5,6] to analyze the information available through a documentary review. We reviewed our databases, the health projects designed by the schools, the annual rapports sent, and the notes a public health physician took during the assessment visits. We followed the standards for evaluation research [7].

To extract information from the projects, rapports, and notes review, we used a data collection sheet designed by us with closed-ended and open-ended questions, inspired by the Innov8 Equity model [8], the SHE recommendations [9], and the quality criteria by Aragon Health promotion Network [10]. We tested the reliability of the sheet, making a double collection with three cases. A public health physician, new in the program, did the documentary review between September 2018 and February 2019.

The information collected was:

Structure variables: level of education (infant, primary, secondary, or adult education); school years in the program; the number of situation analyses performed; the number of school years in which any teacher participated in CEPS training; the number of awards received up to 2019–2020; the number of certificates received.

Process variables: commitment to leave no one behind and prioritization of special populations (equity perspective); advocacy actions favorable to health promotion; coordination with other programs in the school; use of the PHD programs of healthy food and physical activity [11], addiction prevention [12], sexual and affective education [13], or others; use of the online questionnaire on health food and physical activity [11] or others; critical analysis of health problems: relationship with health services and activities developed by health professionals: having a health professional in the health committee, informal seminars, training of teachers in emergencies, health consultancy, or others.

Results variables: quality participation of the educational community, that is: students, teachers, non-teaching staff, families, and external agents (health and social services, NGOs, police, others), according to the classification proposed by White: none, nominal, instrumental, representative, or transformative [14]; quality of the health project according to the HEPS Inventory Tool [15], that includes 37 items, grouped in four aspects: concept, with 17 items; structure, with seven items; process, with six items; and results, with seven items. Each of these aspects could score 0 (low quality), 1 (medium quality), or 2 (high quality). In the end, the sum of these four aspects constituted the final score, qualifying the project as low quality (0–3 points), medium quality (4–5), or high quality (6–8). Finally, we collected the health projects’ results in health, education, social harmony, and democracy.

The data collection sheet also included open-ended questions related to process and results: What was the influence of gender relations? How were the needs of specific special populations evaluated? Which actions have improved the physical environment? Which actions have improved the social environment? Which external agents were part of the project, and what do they do? How did the health project contribute to creating or activating synergies with other community resources? How did the health project contribute to a community organization?

We used a descriptive analysis using relative frequencies and Chi2 contingency analysis for qualitative variables in quantitative data (closed-ended questions). For the qualitative information (open-ended questions and notes from the assessment visits), we performed a content analysis using the equity perspective proposed by the Innov8 [8] and the Ottawa Charter for Health promotion [4] as a theoretical framework.

## 3. Results

Between the 2014–2015 and 2019–2020 school years, 84 schools participated in the CEPS Program, 61 from Mallorca, six from Menorca, 16 from Ibiza, and one from Formentera. The number of participating schools increased each year, from 25 in 2014–2015 to 62 in 2019–2020.

### 3.1. Structure Evaluation

Table 1 presents the characteristics of the schools, comparing the schools included later in the process and results evaluation (only 44 schools: 36 from Mallorca, 6 from Ibiza, 1 from Menorca, and 1 from Formentera) and the ones not included. Adolescents were the most targeted population. There was a higher percentage of public schools included in the process and results evaluation. One out of five schools (24%) that participated in CEPS left the program. In this regard, 86.6% of the schools were left in 2014–2015 and 2015–2016. Most schools included in the process and results evaluation have performed more than one situation analysis, participated in CEPS training, and received a certificate and a grant.

### 3.2. Process Evaluation

Figure 1 shows the model for CEPS development that emerged from the qualitative analysis. Training and other benefits of the program helped the schools understand that the health promotion model required the participation of the entire educational community. Therefore, community participation emerged as a central element for the development of the CEPS Program, linked not only to the participation of the educational community but also to its social context and physical environment. Finally, health services can contribute to CEPS development.

#### 3.2.1. Program Requirements and Benefits

CEPS perceived that the CEP Program was very demanding in terms of paperwork despite the few compensations and recognitions it offers. They have difficulties especially in fulfilling indicators of evaluation, even though some schools sense that qualitative methods could also be useful for the evaluation.

*We’ve talked about the problems that arise when making the health project, especially regarding the annual report. There are many documents to fulfill, and we don’t know how to do it. Just a simple evaluation report on the activities, the review, and improvement proposals would be enough*.[School 7]

As Table 2 shows, CEPS made little use of the health education programs of the PHD (sexual and affective education, prevention of addictions) or the questionnaires (healthy food, physical activity) that are available online. They expressed training needs in sexual and affective education, emotional aspects, and some specific diseases, such as diabetes, epilepsy, and allergies, when they had students with those.

#### 3.2.2. Comprehension of the CEPS Model

Most CEPS have a larger and global vision of health, as they have integrated it in their mission and vision. They understand that health promotion requires time.

*Due to the continuity of many activities, the health project has been established in the life of our educational community as a hallmark of the school*.[School 5]

The coordination of the health committee with the committees of other school programs (such as the environmental or coeducation ones) was a crucial element for developing the CEPS Program. As Table 2 shows, the health committees interacted with the committees of other programs, with different approaches, in more than half of the schools (61.4%).

*The school already had an environmental project, with a vegetable garden, as well as a social harmony committee with the participation of parents, students, and the management team. They have incorporated activities from the environmental project and the social harmony plan to the health project*.[School 21]

*In 2017–2018, we decided to join the health committee with the environmental committee*.[School 31]

Only one out of four schools (27.3%) had a specific objective towards equity and only one out of three gave priority to any special population (Table 2). Equity involves the key concepts of inclusion and equality. On the one hand, inclusion is related to students with economic difficulties or with special needs. On the other hand, equality is related to gender and sexual orientation. Schools appraised those programs that were addressed to distribute fruits or to open public sports facilities because these gave access to healthy products or activities to students with economic difficulties.

*One of the objectives of the project is to improve social harmony, so the health committee has decided to work around equality and gender violence all the school year*.[School 29]

*We need to increase the distribution of fruits to ensure that students with economic difficulties can have them at least once per week. Many of them would not have access to sports facilities if it were not for the ones planned in the project*.[School 10]

#### 3.2.3. Educational Community Participation

The participation of teachers was representative or transformative in all cases, surpassing the participation of the other agents (Figure 2). Involvement of the management team, the continuity of the health coordinator, and the support from the faculty of the high school were strategic elements for teacher participation.

Regarding students, their participation was representative or transformative in half of the cases. They participated by organizing or proposing activities. Furthermore, in some schools, students were part of the health committee, and in others, they organized class delegate meetings.

*Through the class delegate meetings, students have prepared food rules for the parties at the center*.[School 24]

In contrast, the participation of families, non-teaching staff, and external agents was representative or transformative in a lower percentage. Some schools achieved the involvement of families; food seemed to be key for that. In the same way, people in charge of the kitchen or the bar of the center were important for health projects.

*The health committee only wanted fruit juices for the parties. They asked for collaboration to the families, and they brought so many oranges that they didn’t fit in one room. It was a complete success*.[School 22]

*We included the person in charge of the kitchen in the health committee so we can have a more fluid relationship in this important part of our project: the school cafeteria*.[School 9]

#### 3.2.4. Community Participation

There were some interesting cases of community participation through local health committees that were initiatives of the school, the health center, or the town hall. The benefits of these local health committees were multiple, as they allowed contact with different local resources, the mobilization of families through family’s associations, and getting different kinds of aid.

*With the local Committee of Education, Health and Social services, all activities are offered to all schools of the Town Hall. It coordinates all planification and programming. The Town Hall supports the initiatives and searches for the required solutions. The parents’ associations are present, so the activities arrive to every family*.[School 23]

#### 3.2.5. Social Context

Most CEPS made an analysis of the neighborhood. The social context was marked by the socioeconomic situation of the families as well as by immigration and seasonal work. Specific problems with Gypsy populations were observed. Potential for conflict occurred with some students the first year of secondary education, sometimes because of mental health problems. In this complex context, schools tried to adapt themselves.

*We decided to start with emotional awareness to improve the social harmony in this diverse environment… It is vital to give tools to our students to make them aware of their situation, their feelings, etc. As they only perceive their family situation, they could believe that only this reality exists*.[School 22]

#### 3.2.6. Physical Environment

The playground, the bar, and the entrance of the school were the strategic elements of the physical environment of the centers. Many of them had a vegetable garden and they also developed active playgrounds, in which physical activity was promoted. Some schools had arranged with the person in charge of the bar to have fruits and healthy sandwiches, as well as the reduction or elimination of sugary drinks and industrial pastries. Regarding the entrance, biking to school could be useful if there is a bike parking area available in the school. Furthermore, people (be it students, families, or teachers) smoking in the entrance could become a problem. For this reason, the presence of a school police officer that collaborates in terms of conflict resolution emerged as a key agent.

*We did a zumba activity, which was useful not only for promoting physical activity, but also for developing a sense of belonging to a community*.[School 21]

*The school police officer is highly active. Nobody smokes in the school entrance; he takes care of it. It works*.[School 31]

#### 3.2.7. Relationship with Health Services

Most CEPS had a relationship with the health center and other health services, such as the hospital. The relationship with the health center was often from the beginning, as in some cases, it was the health center that encouraged the school to participate in the CEPS Program. *Alerta escolar*, the health center’s program to train teachers in some medical emergencies, received a good evaluation. However, its bureaucracy sometimes delayed the entrance to the program.

### 3.3. Results Evaluation

As is shown in Table 2, 40.9% of CEPS had health results in relation to healthy food and physical activity, and one out of four had results in social harmony. However, the presentation of educational or democracy results in health projects was unusual.

According to the HEPS Inventory Tool [13], 40.9% of CEPS schools obtained a high quality score, while 50% and 9.1% of them obtained a medium quality and a low quality score, respectively.

Factors associated with high quality projects were: the obtaining of a grant from the PHD (*p* = 0.029), the transformative participation of teachers (*p* = 0.029) or, at least, the instrumental participation of external agents (*p* = 0.004), and the presentation of health results (*p* = 0.001) (Appendix A).

## 4. Discussion

It was optimistic that the number of schools in the CEPS Program increased every year and that the attrition rate was low. The program’s attrition occurred during the first two years of its development and, mainly, during the first two years of the school being in the program. Perhaps, it was due to the lack of training and assessment in the schools during the first two school years of the program, elements identified in other studies as critical for implementing health promoting schools’ programs [16,17]. Another factor could have been the change of the person responsible for the school’s program, especially if the CEPS Program was not part of the Educational Plan of the Centre. We believe that mandatory initial training of teachers in health promotion and further support and counselling to the schools during the beginning of the program could help prevent them from leaving the program.

Most CEPS made no use of the PHD Health Education Programs, based on the training of teachers in different health topics. CEPS and PHD Health Education Programs required effort from teachers. It was easier for the schools to request an informal seminar from the health center or another external agent from the nurse. Informal seminars were the most demanded activity by the CEPS schools of the health centers. The problem was that they were isolated activities and could have been a complement, but not the core. For this reason, it would have been beneficial if health professionals from the health centers and other external agents knew more about the CEPS Program and the PHD Health Education Programs and recommended it to the schools. Nevertheless, the IECD should encourage additional teachers’ training, while the PHD should increase its training capacity, which is currently limited, and at the same time develop a marketing strategy similar to that of other healthcare administrations [18]. Innovative concepts such as add-in (health-promoting activities that become part of curriculum-based educational activities without taking time away from core curriculum obligations) [19] or the intergenerational learning (from children to parents) [20] are promising. They could help the development and sustainability of the health-promoting school programs.

Most CEPS understood the health promotion model and the need for enough time to develop it. At regional and national levels, the coordination between Health and Education administrations and the continuous commitment to School Health Promotion independently of the political party in Government was essential.

One of the barriers identified for the development of the CEPS Program was the excessive bureaucracy. This is a shared problem with other Spanish health promoting schools [21]. We are working at making the CEPS Program simpler and more flexible. In parallel, the Spanish Ministry of Health, in collaboration with the Spanish regions, is writing a National Guidelines for Health Promoting Schools, that aims to harmonize these programs.

The evaluation of the health projects was another difficulty for the CEPS. Therefore, in the PHD, we built EinaSalut (www.einasalut.caib.es (accessed on 3 July 2022)), a website platform for health literacy inspired by One You, now called Better Health (https://www.nhs.uk/better-health/ (accessed on 3 July 2022)), with a specific site for the CEPS Program. At EinaSalut, we offer different instruments to help CEPS evaluate their health project, such as the Health Behavior in School-aged Children questionnaire [22], based on our previous experience with online questionnaires regarding healthy food and physical activities [11]. On this page, we encourage schools to use mixed designs, combining quantitative techniques with qualitative methods, following the model we used in this study. At the same time, we observed that CEPS focus the evaluation of health projects on health results, mainly on healthy food and physical activity. As there is evidence that School Health Promotion programs could also obtain educational or well-being results [23], we should encourage CEPS to also use these indicators to evaluate their health projects.

The coordination of CEPS with other schools’ programs helped its development and improved its quality and that of the other programs. Schools know that working on the environment, social harmony, or equality involves working on health. In this sense, the frame of the Sustainable Development Goals [24] could be helpful for schools, so IECD and PHD should promote their use. In the last years, the IECD announced all the programs in the same call, reinforcing the link between the programs.

Family and community engagements are two challenging and complex elements for developing the CEPS Program and, generally, all the Health Promotion programs at school [25]. Nevertheless, we observed that family engagement is possible, especially when linked to healthy food activities and the neighborhood promoted it by offering spaces for the coordination between schools, families, and other local organizations, optimizing the available resources and taking advantage of every opportunity. These local health platforms emerged as a critical element for community engagement, not only for the CEPS Program but also for the so-called Family Health promotion [26].

Most of the CEPS performed an exhaustive analysis of their social context; however, this was not taken into account when they designed the health project. There is evidence that tailoring the health projects to the individual schools’ needs is a critical element for their success [25] and an essential component to introduce equity in them. We used the Innov8 tool [8] to review the incorporation of equity in the CEPS Program, and we observed that it is a pending subject. The Spanish Ministry of Health has adapted the Innov8 questionnaire for use in all health strategies, programs, or activities [27]. We propose their use for the CEPS health projects as it offers many opportunities for improvement and critical thought and because the path to equity in schools is linked not only to the concept of health literacy [28] but also to mental health literacy [29]. Indeed, the CEPS Program should add mental health as a specific thematic area.

Participation of the educational community also required the participation of students and non-teaching staff. Students desired more involvement in the educational community and considered that health promotion activities related to physical environment and project-based within the curriculum had the maximum potential [30]. Youth engagement presented an exciting prospect for CEPS and other health promoting schools to gain a deeper understanding of the factors that influence individual and community health and wellbeing, to help to develop responsive policies and health projects [31]. Regarding non-teaching staff, as canteens managers, they required training in health food [32]. From our point of view, training in CEPS Program will also be suitable.

Limitations of the evaluation: The main limitation is intrinsic to the method, as we have used only written information. We also planned to visit the CEPS to interview the health committees and make a participant observation of the surrounding areas, entrance, playground, bar, or spaces. Unfortunately, visits were suspended in March 2020 because of the COVID-19 pandemic, preventing us from triangulating the written information with the educational community’s live account or direct observation. Nevertheless, we had the field notes from the PHD physician who had visited the schools in the previous school years. We hope these visits could be possible in the next school year, as the direct voice of teachers, families, and students is indispensable for improving the CEPS Program.

## 5. Conclusions

Most CEPS have high or medium quality health projects. After two years in the program, schools do not abandon it. Coordination with other programs at school and with other schools and families through local health committees and the coordination between the Health and Education administrations are critical elements for the development of the CEPS Program.

## Figures and Tables

**Figure 1 ijerph-19-10756-f001:**
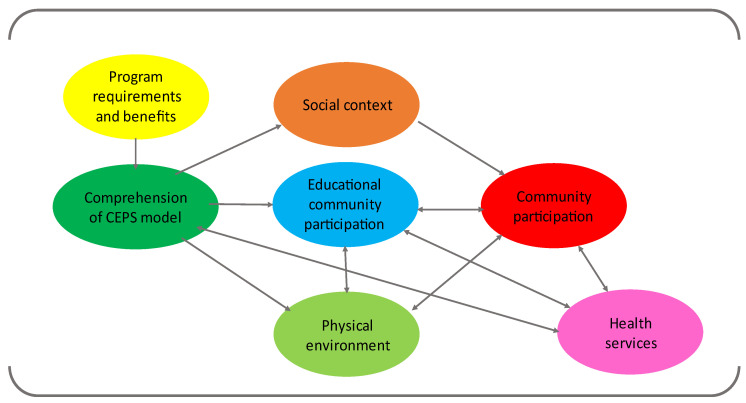
Model for development of CEPS.

**Figure 2 ijerph-19-10756-f002:**
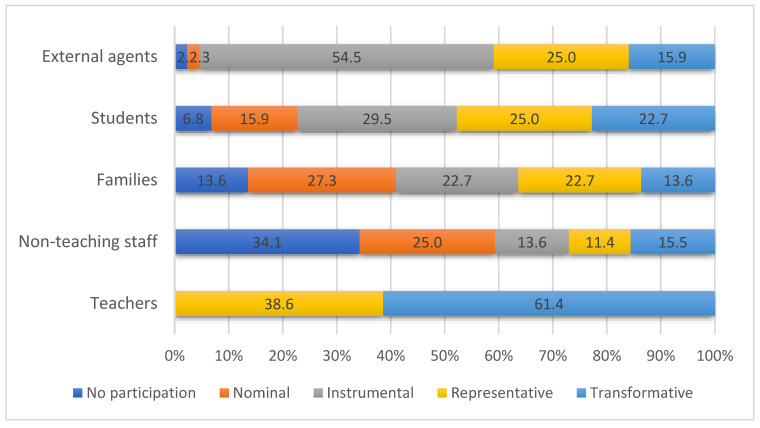
Educational community participation [14] in the CEPS Program.

**Table 1 ijerph-19-10756-t001:** The schools′ characteristics according to the evaluation of the CEPS Program.

Variable	Categories	Only Structure	Structure, Process & Outcomes	TOTAL
Type of school	Infant *	2 (5.0)	5 (11.4)	7 (8.3)
	Infant + Primary	1 (2.5)	0 (0.0)	1 (1.2)
	Secondary	9 (22.5)	11 (25.0)	20 (23.8)
	Adults	16 (40.0)	21 (47.7)	37 (44.0)
	Infant + Primary + Secondary *	10 (25.0)	5 (11.4)	15 (17.9)
	Total	40 (100.0)	44 (100.0)	84 (100.0)
Years participating in the CEPS Program	1	31 (75.0)	0 (0.0)	31 (36.9)
	2	9 (22.5)	3 (6.8)	12 (14.3)
	3	0 (0.0)	13 (29.5)	13 (15.5)
	4	0 (0.0)	2 (4.5)	2 (2.4)
	5	0 (0.0)	12 (27.3)	12 (14.3)
	6	0 (0.0)	14 (31.8)	14 (16.7)
	Total	40 (100.0)	44 (100.0)	84 (100.0)
Number of situation analysis performed	0	8 (21.6)	1 (2.6)	9 (10.7)
	1	21 (56.8)	9 (23.7)	30 (35.7)
	2	7 (18.9)	18 (47.4)	25 (29.8)
	3	1 (2.7)	7 (18.4)	8 (9.5)
	4	0 (100.0)	3 (7.9)	3 (3.6)
	Total	40 (100.0)	44 (100.0)	84 (100.0)
Participation of any teacher in CEPS training	0	18 (45.0)	4 (9.1)	22 (26.2)
	1	18 (45.0)	14 (31.8)	32 (38.19
	2	4 (10.0)	13 (29.5)	17 (20.2)
	3	0 (0.0)	8 (18.2)	8 (9.5)
	4	0 (0.0)	5 (11.4)	5 (6.0)
	Total	40 (100.0)	44 (100.0)	84 (100.0)
CEPS grant	0	33 (82.5)	19 (43.2)	52 (61.9)
	1	4 (10.0)	12 (27.3)	16 (19.0)
	2	3 (7.5)	6 (13.6)	9 (10.7)
	3	0 (0.0)	7 (15.9)	7 (8.3)
	Total	40 (100.0)	44 (100.0)	84 (100.0)
CEPS certificate	Yes	0 (0.0)	30 (68.2)	30 (35.7)
	No	40 (100.0)	14 (31.8)	54 (64.3)
	Total	40 (100.0)	44 (100.0)	84 (100.0)

* Subsidized schools. The other categories correspond to public schools.

**Table 2 ijerph-19-10756-t002:** Process and results evaluation of the CEPS Program (N = 44) *.

Process	Number	%
Commitment to leave no one behind	12	27.3
Prioritization of any special population	14	31.8
Coordination with other programs	27	61.4
Advocacy actions favorable to health promotion	23	52.3
Use of PHD programs & questionnaires	Number	%
- *Bon dia salut* (abilities for life)	2	4.5
- *Decideix* (prevention of addictions)	6	13.6
- *THC supera el repte* (prevention of cannabis consumption)	2	4.5
- *Respiraire* (prevention of tobacco consumption)	5	11.4
- *Amb tots els sentits* (sexual & affective education)	3	6.8
- *Sexe segur i responsable* (prevention of sexually transmitted diseases)	5	11.4
- Other programs (prevention of addictions; sexual education)	9	20.5
- *Questionaris dieta mediterrània/vida activa* (healthy food/physical activity)	6	14
- Other questionnaires (healthy food & physical activity; general health)	20	45.5
Process	Number	%
Critical analysis of health problems	20	45.5
Relationship with health center	36	81.8
Relationship with other health services	12	27.3
Activities performed by health professionals:	20	45.5
- Health professional in the health committee	12	27.3
- Informal seminars	23	52.3
- Alerta escolar (training of teachers in health emergencies)	14	31.8
- Consulta jove (health consultancy at school)	19	43.2
- Other activities	14	31.8
Results	Number	%
Health	18	40.9
Education	2	4.5
Social harmony	12	27.3
Democracy	1	2.3
Total schools	44	100

* Multiple choices items.

## Data Availability

Not applicable.

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
