# Peer review of "Evaluation of the Health Promoting Schools (CEPS) Program in the Balearic Islands, Spain"

_ijerph, 2022, doi:10.3390/ijerph191710756_

Round 1

Reviewer 1 Report

Brief Summary

The manuscript presents an evaluation of the structure, process and results achieved by the CEPS Programme in the Balearic Islands. In particular, the Authors identify the elements that have helped or made the implementation of the Programme difficult. Based on their findings, the Authors develop recommendations that could help both the schools and the PHD and IECD to improve.

The authors base their analysis on a document review. To extract information from the projects, reports and reviews, the Authors develop a data collection form with closed and open-ended questions.

The manuscript illustrates a descriptive analysis based on relative frequencies and Chi2 contingency analysis for quantitative data (closed-ended questions). In addition, the manuscript illustrates a content analysis that follows the equity perspective proposed by Innov8 and the Ottawa Charter for Health Promotion as a theoretical framework.

Based on the results of the analysis, the authors discuss how the training and other benefits of the programme helped schools realise that the health promotion model requires the participation of the entire educational community. Thus, community participation emerges as central to the development of the CEPS Programme. 

Finally, the authors discuss several research issues: the identification of a certain degree of attrition and its causes; the lack of specific training on the CEPS Programme and PHD health education programmes; the implementation of the EinaSalut digital platform and other facilitating factors (coordination of CEPS with other school programmes, involvement of families, neighbourhood analysis); research limitations.

General comments 

The article is well written and illustrates an interesting qualitative-quantitative research conducted in the field. Apart from some minor remarks (see below), the main weakness of the manuscript seems to be the limited number of references and their relative obsolescence. In particular, only 22 references are cited and most of them were published more than 5 years ago. 

Additional literature could facilitate the Authors in revising Section 3 (Results). In particular, while Figure 1 provides a clear picture of the evaluation process, many topics deserve a more accurate illustration. In particular, community participation, the social context, the physical environment and the relationship with health services might deserve a more in-depth analysis, but the other topics could also be better explained. 

Furthermore, the data illustrated in Table 2 could be presented using histograms or bar graphs, and the authors could illustrate in a few lines why the relative frequencies do not reach 100% (clearly, multiple answers were allowed).

Moreover, in the discussion section, the illustration of two or three similar cases of educational communities can enrich the interpretation of the results obtained and help to improve and update the bibliography. 

Finally, a few lines of more detailed explanation of the tests associated with the p-values presented in Table 3 (briefly mentioned in line 109) could facilitate the reader in understanding the illustrated results.   

Minor comments

The authors are requested to check Table 1, as it appears that two lines are missing in its first part. If this is not the case, the authors are invited to add a few lines in the main text to explain the particular structure of the table.

The authors are invited to move Table 3 to a dedicated appendix to improve the readability of the text. 

Author Response

Thanks for your review. Our responses are in the file attached.

Reviewer 2 Report

Thank you for the opportunity to learn about this health promotion project in schools. It is good to see initiatives to improve especially child and adolescent health. Regarding the manuscript, there are several details that I recommend for your review. Also, some questions are raised in this regard.

- Line 2: There is a typo. Promotion or Promoting?
- Line 4: Names should appear first without abbreviations, followed by surnames. The penultimate number should be in superscript and the comma at the end should be eliminated.
- Line 9: This is the objective, so the verb should be in the infinitive form.
- Line 25: (1) In the Methodology, the Ottawa Charter is mentioned as a theoretical framework, but it is not explained in the Introduction. (2) On the other hand, are health issues dealt with in the different subjects of the educational centres?
- Line 26: Why did this program come about? What were the factors that influenced its creation? Have similar programs been carried out in other regions of the country?
- Line 38: Ibiza? The word is written in Catalan. Check this detail in the rest of the manuscript, as it is repeated on some occasions.
- Line 40-41: What does adult education mean? Do you mean the university that exists in the Balearic Islands?
- Line 49: Please include the comma in the thousands (2,000).
- Lines 63-64: The comment on quantitative and qualitative methods is a bit unspecific, could you please add more information to complement what is written in lines 64-67?
- Lines 74-75: This text appears similarly in lines 61-62, so there is no need to repeat the information.
- Lines 109-110: Although I understand the intent of the wording, it should be added that chi-square will be used for qualitative variables in quantitative data.
- Lines 114-116: This text is not necessary, it could be deleted.
- Table 1: The first part of the table does not present the last two columns.
- Line 133: Regarding this whole section, qualitative results from reports from schools are included, but sometimes it is confusing to know if the sentence is from the authors or if they are exact texts from the reports. Use italics where appropriate.
- Lines 149-150: The expression "we don't know how to do it", which I understand to be one of the main limitations of this project, is not addressed later in the discussion.
- Table 2: (1) It is convenient to improve the format of the table to better differentiate the sections and subsections that exist in it. (2) Some words are in Catalan, so it is recommended to use italics in these cases.
- Line 243: "Alerta escolar" are words in Catalan. Please use italics.
- Table 3: Again, there are Catalan words that should be italicized.
- Lines 258-260: If you need to indicate the source of the questionnaire, it can be simplified to: "Questionnaire used: HEPS Inventory Tool [13]".
- Lines 272-274: Why are these programs not mandatory? It is assumed that, for this project to be successful, the teachers participating in the project should first be trained in knowledge and skills.
- Lines 289-290: It seems that this sentence is incomplete. Perhaps some information should be added at the beginning of the sentence.
- Lines 320-325: (1) The limitations do not correspond to the project itself. Part of the limitations of the project itself can be extracted from the school reports. For example, it has been found that teachers have problems filling in documents. Also, there may be a possible loss of data in the transfer of information from the reports to the analysis sheets. (2) If there are field notes, why have they not been used for the analysis of the project? It is understood that the qualitative data provided are from the school reports.
- Lines 332-348: Please delete the text that corresponds exclusively to the template.
- Line 349: The bibliography is not adapted to the template provided by IJERPH.

I hope the comments will be helpful to improve the manuscript.
Best regards.

Author Response

Thank you for your review. The responses are in the file attached.

Round 2

Reviewer 2 Report

Thank you for the thorough revision of the contents of the original manuscript. Clearly, there has been a positive change, which reduces the number of new suggestions:

- Line 28: I am sorry for the difficulties in understanding one of the questions. It was related to the fact that, in many countries, there are subjects that incorporate health-related content in their curriculums. Does the programme rely on this content to develop its activity?

- Line 39: It has not been explained what the Ottawa Charter implies in relation to what was done in the study. Also, please change "Otawa" to "Ottawa".

- Line 108: After review of this section, I think the heading of this section ("Analysis:") should be removed. This is because the rest of the content of this section is not divided into sections.

- Table 1: (1) Please replace the colon in the heading with a full stop. The same comment applies to Table 2. (2) Some results should be revised with respect to the totals. For example, the sum in the categories "Infant + Primary", "Secondary", and "Adults" does not correspond to their total. Please check the rest of the sums in the table, as the same problem appears in other occasions.

- References: Some improvements are needed. In articles, journal names and volumes should be in italics and years in bold.

I hope these suggestions can improve the manuscript.

Best regards.

Author Response

Thanks again for your comment. I have made the changes suggested.
